# Bridging Science and Practice on Multi-Hazard Risk Drivers: Stakeholder Insights from Five Pilot Studies in Europe

Nicole van Maanen[1], Marleen de Ruiter[1], Wiebke Jäger[1], Veronica Casartelli[2], Roxana Ciurean[3], Noemi Padrón-Fumero[7], Anne Sophie Daloz[4], David Geurts[5], Stefania Gottardo[2], Stefan Hochrainer-Stigler[6], Abel López Diez[7], Jaime Díaz Pacheco[7], Pedro Dorta Antequera[7], Tamara Febles Arévalo[7], Sara García González[7], Raúl Hernández-Martín[7], Carmen Alvarez-Albelo[7], Juan José Diaz-Hernandez[7], Lin Ma[4], Letizia Monteleone[2], Karina Reiter[6], Tristian Stolte[1], Robert Šakić Trogrlić[6], Silvia Torresan[2], Sharon Tatman[5], David Romero Manrique de Lara[7], Yeray Hernández González[7] and Philip J. Ward[1,5]

Corresponding author: Nicole van Maanen (n.a.m.van.maanen@vu.nl)

[1] Institute for Environmental Studies, Vrije Universiteit Amsterdam, the Netherlands
[2] CMCC Foundation - Euro-Mediterranean Center on Climate Change, Venice, Italy
[3] British Geological Survey, Keyworth NG12 5GG, United Kingdom
[4] Center for International Climate Research, Oslo, Norway
[5] Deltares, Delft, the Netherlands
[6] IIASA - International Institute for Applied Systems Analysis, Laxenburg, Austria
[7] Universidad de La Laguna, Spain

## Abstract

Effective disaster risk management requires approaches that account for multiple interacting hazards, dynamic vulnerabilities, and institutional complexity. Yet many existing risk assessment methods struggle to reflect how these risks evolve in practice. This paper explores multi-hazard risk dynamics through stakeholder interviews across five European regions (Veneto, Scandinavia, the North Sea, the Danube Region, and the Canary Islands). Stakeholders described how exposure and vulnerability shift over time due to climate change, urban development, and socio-economic dependencies. The interviews highlight governance challenges and the critical role of institutional coordination, as well as synergies and asynergies in DRR measures, where efforts to reduce one risk can unintentionally increase another. By foregrounding real-world experiences across diverse hazard landscapes and sectors, this study offers empirical insights into how multi-hazard risk is perceived and managed. It underscores the need for flexible, context-sensitive strategies that bridge scientific assessment with decision-making on the ground.

## 1. Introduction

Risks are increasing globally, driven by climate change, environmental degradation, and socio-economic transformations, among other factors and processes (e.g., IPCC, 2022; CRED, 2021; Poljanšek et al., 2017). The complexity of disaster risk is further amplified by the interplay of multiple hazards, which may occur simultaneously, sequentially, or through cascading effects (van den Hurk et al., 2023; Simpson et al., 2021). Recognizing these interconnections, disaster risk reduction (DRR) frameworks have increasingly emphasized the need for a multi-hazard, systemic approach to risk assessment and management that captures the dynamic interplay between hazards, vulnerabilities, and socio-economic processes (Hochrainer-Stigler et al., 2023; Simpson et al., 2021). While DRR focuses on long-term efforts to prevent new risks and reduce existing ones, disaster risk management (DRM) encompasses the broader cycle of preparedness, response, and recovery. Both domains are now converging around the need for integrative approaches that address cascading effects, cross-sectoral interdependencies, and evolving system dynamics.

This shift is also reflected in global policy agendas. The Sendai Framework for Disaster Risk Reduction (UNDRR, 2015) and its Midterm Review (UNDRR, 2023) highlight the importance of understanding the dynamics of risk and its underlying drivers, stressing that while progress

has been made in risk reduction, new risks continue to emerge and accumulate at a faster pace than they are reduced. Fragmented governance structures, sectoral silos, and a lack of systemic foresight contribute to the persistence of vulnerabilities and the creation of new risks, underscoring the urgent need for integrated, forward-looking approaches (Allen et al., 2023).

Despite growing recognition of the need for multi-hazard risk assessments, significant challenges remain in translating this understanding into effective practice (Senevirathne et al., 2024; Ward et al., 2022; Poljanšek et al., 2017; Šakić Trogrlić et al., 2024). Various methods have been developed to assess multi-(hazard-) risk, including the use of disaster databases (Jäger et al., 2025; Lee et al., 2024; Delforge et al., 2023), combining single hazard footprints (Claassen et al., 2023), probabilistic risk models (Stalhandske et al., 2024; Zscheischler et al., 2020), and artificial intelligence-driven forecasting techniques (Qin et al., 2024). Earlier foundational contributions to this field include work by Kappes et al. (2012), Gill and Malamud (2014, 2016), Liu et al. (2016), and Tilloy et al. (2019), while broader research agendas and conceptual developments are outlined in Ward et al. (2022). These approaches provide valuable insights into hazard interactions and exposure patterns, yet they often struggle to capture the rich, context-specific information that shapes risk at local and regional scales (Gallina et al., 2016). Disaster databases, while useful for historical analysis, typically lack granularity on cascading impacts and vulnerabilities (Jones et al., 2023). Probabilistic models, though effective for estimating hazard probabilities, often fail to account for the complex feedback loops that characterize multi-hazard environments (Stalhandske et al., 2024). Emerging machine learning techniques offer promising advancements but remain constrained by data biases and limited integration of qualitative insights from local communities and stakeholders (Albahri et al., 2024). Furthermore, evidence suggests that purely data-driven approaches often overlook societal inequalities when designing DRR measures (Haer and de Ruiter, 2024).

To address these gaps, the MYRIAD-EU project applies an integrated approach to multi-hazard risk assessment, combining quantitative analysis with qualitative insights from stakeholders across five diverse European pilot regions: Veneto (north-eastern Italian region), Scandinavia, the North Sea, the Danube Region, and the Canary Islands (Ward et al., 2022). These regions represent distinct hazard landscapes and socio-economic contexts, spanning critical sectors such as energy, food and agriculture, tourism, ecosystems and forestry, infrastructure and transport, and finance (additionally, water was added as a sector of interest). By examining hazard combinations, vulnerability characteristics, and disaster risk reduction measures within and across sectors, the project aims to provide a more comprehensive understanding of multi-hazard risk dynamics alongside practical tools, methods, and frameworks for developing disaster risk management pathways.

However, a significant gap remains in understanding how risk evolves over time, largely due to the limited integration of dynamic vulnerability and local stakeholder perspectives into existing methodologies (de Ruiter & van Loon, 2022; Gill & Malamud, 2016). This is especially problematic given fragmented governance structures, which hinder the coordination of risk reduction efforts across sectors and scales (Šakić Trogrlić et al., 2024). Incorporating stakeholder insights is therefore not only vital for capturing locally grounded knowledge of cascading impacts, but also for informing more integrated and adaptive governance strategies (de Ruiter & van Loon, 2022; Šakić Trogrlić et al., 2024).

The primary objective of this study is to advance the understanding of multi-hazard risk dynamics by integrating diverse perspectives and identifying key barriers and opportunities for improving risk governance. A central component of this research involves semi-structured interviews with stakeholders within the MYRIAD-EU project, including policymakers, emergency managers, energy operators, and civil protection agencies. These interviews are

structured around four core themes: hazard combinations, vulnerability characteristics, changes in exposure and vulnerability over time, and the synergies and trade-offs of DRR measures. Given the exploratory, qualitative design of the interview study, the findings presented below should be understood as illustrative rather than statistically generalizable.

By capturing qualitative narratives of risk, these interviews provide valuable insights into how stakeholders perceive the effectiveness of DRR measures in addressing multi-hazard and multi-risk scenarios, as well as the synergies and asynergies of these measures across different sectors. While existing methodologies often emphasize quantitative risk modeling, this study highlights the importance of integrating local perspectives and real-world decision-making processes – both of which are essential for developing actionable and context-sensitive DRR strategies (Hermans et al., 2022; Šakić Trogrlić et al., 2024; Parviainen et al., 2025).

## 2. Methods

This study employed semi-structured interviews to explore the dynamics of multi-hazard risk in five European pilot regions. Semi-structured interviews offer a balance between consistency and openness, enabling comparability across participants while allowing rich, contextual exploration of their experiences and perspectives (Clark et al., 2021; Flick, 2022). This flexibility was essential given the diversity of regional contexts and stakeholder groups engaged in the MYRIAD-EU project.

2.1 Stakeholder engagement and selection

The interviews were embedded in a broader stakeholder engagement strategy developed by the MYRIAD-EU project (Ciurean et al., *in prepapration*). Each pilot region was coordinated by a pilot lead with deep contextual knowledge and well-established stakeholder networks. These five pilot regions address different DRM challenges reflecting their unique hazard profiles and socio-economic contexts. The North Sea pilot focuses on optimizing spatial planning at the interface of land and sea to manage increasing and interconnected risks. The Canary Islands pilot aims to enhance resilience in island regions highly dependent on tourism facing multi-hazard risks. The Scandinavia pilot works on maintaining healthy ecosystems while meeting rising demands for energy, food, and ecosystem services, emphasizing nature-based solutions. The Danube Region pilot targets resilience to multi-hazards impacting several interconnected countries with strong economic ties. Lastly, the Veneto pilot seeks to develop forward-looking multi-risk planning across diverse landscapes, from mountains to the sea. This overview provides essential context for understanding the thematic analysis of stakeholder perspectives across the pilots.

Stakeholders were selected to ensure representation of relevant sectors and viewpoints (Nowell et al., 2017). Selection criteria included stakeholder influence, domain expertise, and their relevance for disaster risk management in the context of multi-hazard risks. Stakeholders were drawn from government, civil protection, private sector, NGOs, and academia. Some participants had previously collaborated with the project, fostering trust and openness during interviews. Table 1 provides an overview of the interviews conducted per pilot region.

Pilot leads coordinated the engagement, ensuring contextually relevant and ethically appropriate processes. Stakeholders were provided with background information, definitions (Box 1), and informed consent forms before the interviews. All procedures followed ethical guidelines, and interview recordings and transcripts were securely stored and anonymized.

*Table 1: Overview of interviews conducted in the context of the underlying study as part of the MYRIAD-EU project.*

| Pilot Region | Number of interviews | Number of interviewees | Stakeholder groups |
|---|---|---|---|
| Veneto | 4 | 6 | Regional government authorities and agencies (including civil protection) |
| Scandinavia | 1 | 5 | Energy operators, infrastructure managers, national government authorities |
| North Sea | 3 | 3 | Offshore energy developers, maritime operators, regulatory authorities |
| Danube Region | 5 | 5 | Basin-wide organizations, international NGO, academia, water management agency |
| Canary Islands | 9 | 25 | Civil protection agencies, tourism boards, environmental NGOs, energy and water operators, farmer co-op |


2.2 Interview design and implementation

Interviews followed a common structure co-designed by the central research team and pilot
leads, drawing on insights from previous methodological work within the project. That task
focused on expert interviews to explore systemic risk feedbacks and interdependencies and
helped inform the thematic structure and framing of the stakeholder interviews. Interview
themes were aligned with key dimensions of multi-hazard risk and risk driver feedbacks:

1. Hazard combinations
2. Vulnerability characteristics
3. Changes in exposure and vulnerability
4. Synergies and asynergies of disaster risk reduction measures

These themes were selected to reflect key dimensions of risk dynamics while enabling
comparison across regions.  Additionally, they provide empirical insight into dynamic
feedbacks in risk drivers, supporting the refinement of methods and tools within the project.
Interviews were conducted both individually and in groups. Group interviews fostered dynamic
discussion and co-reflection among participants, while individual interviews allowed for deeper
exploration of personal or institutional perspectives (Guest et al., 2017). Interviews were
conducted in English or in local languages (Italian, Spanish, and Scandinavian languages)
depending on participant preference and context. Where interviews were held in local
languages, transcripts were translated into English and verified for accuracy by the respective
pilot leads.

Interviews lasted between 45 and 90 minutes and were recorded with prior informed consent.
All interviewees received a description of key concepts (Box 1) to ensure shared
understanding and clarity. The interviews were not intended to generate statistically
representative findings, but rather to screen and illustrate diverse stakeholder experiences
and perspectives on multi-hazard risk and its dynamics. To support a shared starting point for
discussion, interviewees were provided with a short glossary (Box 1) defining key concepts
such as hazard interactions, vulnerability, exposure, and resilience. However, during the
interviews it became clear that stakeholders interpreted these terms in diverse and sometimes
conflicting ways, shaped by their sectoral responsibilities, past experiences, and institutional
cultures. Rather than constraining dialogue, these differences proved analytically valuable,
underscoring the importance of aligning scientific terminology with practice-based knowledge
when addressing multi-hazard risk. This experience highlights that bridging scientific and
practitioner perspectives requires more than standardized definitions; it requires ongoing
dialogue about meaning and application.

| Term | Definition | Source |
|---|---|---|
| Hazard | A process, phenomenon or human activity that may cause loss of life, injury or other health impacts, property damage, social and economic disruption, or environmental degradation. | UNDRR, 2016 |
| Exposure | The situation of people, infrastructure, housing, production capacities and other tangible human assets located in hazard-prone areas. | UNDRR, 2016 |
| Vulnerability | The conditions determined by physical, social, economic, and environmental factors or processes which increase the susceptibility of an individual, a community, assets, or systems to the impacts of hazards | UNDRR, 2016 |
| Disaster risk reduction | Preventing new and reducing existing disaster risk and managing residual risk, all of which contribute to strengthening resilience and therefore to the achievement of sustainable development. | UNDRR, 2016 |
| Synergies | See Asynergies (below) | |
| Asynergies | Potentially unwanted effects of measures that reduce the impacts of disasters across different hazards. Traditionally, those measures are aimed at decreasing the risk [e.g., a building faces] of a single hazard type despite their potential of having unwanted effects on other hazard types. For example, building on stilts is an often-used measure to decrease a building's flood vulnerability, however, it simultaneously increases a building's earthquake vulnerability. | De Ruiter et al., 2020 |
| Amplication effects | The occurrence of one hazard can increase the likelihood and/or magnitude of additional hazards in the future (e.g., forest fires can amplify the triggering of debris flows during heavy rain) | Ciurean et al., 2018 |
| Multi-hazard | The selection of multiple major hazards that the country faces, and the specific contexts where hazardous events may occur simultaneously, cascadingly or cumulatively over time, and considering the potential interrelated effects. | UNDRR, 2016 |
| Multi-risk | Risk generated from multiple hazards and the interrelationships between these hazards (and considering interrelationships on the vulnerability level). | Zschau, 2017 |
| Multi-(hazard-)risk | Risk generated from multiple hazards and the interrelationships between these hazards (but not considering interrelationships on the vulnerability level). | Zschau, 2017 |

*Box 1: Key terms and definitions in multi-hazard risk assessment provided to stakeholders before interviews.*

2.3 Data analysis

A flexible thematic analysis approach was used to examine the interview data, enabling both structure and adaptability in identifying patterns across the pilot regions (Nowell et al., 2017; Miles et al., 2014). The analysis was primarily deductive, guided by the four predefined themes used in the interview design - hazard combinations, vulnerability characteristics, changes in exposure and vulnerability, and synergies and asynergies of disaster risk reduction measures. These themes were derived from the project's conceptual framework and structured the initial stages of coding (Guest et al., 2012; Nowell et al., 2017).

At the same time, the analysis remained open to inductive insights, allowing for the identification of themes and cross-cutting issues that emerged from the data itself. In qualitative research, deductive analysis involves applying a coding framework based on existing theory or predefined research questions, whereas inductive analysis allows patterns and themes to emerge organically from the data without being constrained by prior expectations (Bonner et al., 2021). By combining both approaches, the analysis was able to reflect both consistency across regions and the grounded, contextual experiences of interviewees.

The analysis involved a systematic review of all transcripts, identifying recurring topics, illustrative stakeholder direct quotes, and region-specific dynamics. ATLAS.ti (ATLAS.ti, 2023) was used to support the organization and tagging of transcript segments, but the analytical process itself was primarily interpretive and iterative. It involved multiple cycles of reading, memo-writing, and comparison across interviews to develop and refine codes, following best practices for qualitative thematic analysis (Williams & Moser, 2019). The initial coding structure was aligned with the four core interview themes, while additional codes were developed inductively to capture emerging issues (i.e., governance fragmentation) that cut across most regions and sectors.

Pilot leads played a crucial role in interpreting and contextualizing findings, particularly for interviews conducted in local languages. They verified translation accuracy, explained institutional arrangements, and contributed to the identification of plausible interpretations in line with local governance and environmental contexts. This collaborative, multi-actor analysis process reflects principles of researcher triangulation, helping to reduce individual bias and enhance the credibility and confirmability of findings (Nowell et al., 2017).

The coding process followed an iterative two-stage structure. First, a deductive coding frame was applied based on the four predefined interview themes (hazard combinations, vulnerability characteristics, changes in exposure and vulnerability, and synergies and asynergies of DRR measures). In the second stage, additional inductive codes were developed to capture emergent issues raised across interviews, such as governance fragmentation and resource constraints. Codes were refined through several rounds of review and discussion among the analytical team to consolidate and adjust overlapping categories. While ATLAS.ti supported data organization and tagging, the interpretation of meaning remained an iterative, manual, and reflective process. This level of transparency is intended to enable readers to understand the analytical logic and ensure methodological reproducibility without requiring access to internal code files.

To further ensure trustworthiness, selected quotes were shared with interviewees for validation prior to publication. This provided an opportunity for participants to confirm the accuracy and intent of how their perspectives were represented.

The findings are presented thematically in Section 3 and illustrated with direct stakeholder quotes to convey depth and diversity of experience. The analysis should be understood as illustrative rather than exhaustive - designed to surface key insights and dynamics rather than provide a systematic or statistically generalizable assessment. By combining structured thematic review with grounded interpretation, the approach offers a meaningful synthesis of stakeholder perspectives across diverse regional settings.

**3. Results**

This section presents key findings of the thematic analysis of data collected in the pilot regions, focusing on hazard interactions (3.1), vulnerability characteristics (3.2), and changes in exposure and vulnerability characteristics (3.3). It also examines the effectiveness of DRR measures, highlighting both synergies and asynergies (3.4). It is important to note that the scale of the pilot regions varies significantly, from relatively small and concentrated areas like Veneto and the Canary Islands, to much larger, multi-national regions such as the Danube Region, Scandinavia, and the North Sea, and the analysis is not intended to be exhaustive; rather, it aims to provide illustrative examples of key interactions and dynamics.

3.1 Hazard combinations

The interactions between multiple hazards vary across the pilot regions, often exacerbating
disaster risks and amplifying cascading and compounding effects. Here, we present some
combinations that were highlighted during the interviews.
In Veneto, intense rainfall often triggers flooding and landslides, especially in urban and
mountainous areas, while storm surges are intensified by high tides and southern winds,
increasing coastal flood risk. These insights reflect the perspectives of regional civil protection
authorities, municipal disaster managers, and forestry and industrial sector representatives,
who emphasized the operational and environmental implications of cascading flood and
landslide events. Events like Storm Vaia (2018) have caused cascading impacts, including
forest loss, and avalanches (Casartelli et al., 2025). Flooding in industrial zones also raises
concerns about chemical contamination. In Scandinavia, heavy precipitation combined with
storms and higher temperatures increases the likelihood of flooding and landslides,
particularly in late winter and early spring when snowmelt saturates soils. These perspectives
reflect the experiences of regional emergency managers, infrastructure operators, and land-
and energy-sector producers, who highlighted operational challenges associated with rapidly
shifting climatic conditions. Interviewees noted that these compound events are becoming
more unpredictable and difficult to manage. As one interviewee explained, "*One of the big*
*challenges is the quick changes. Many of the producers are not adjustable, so they need to*
*prepare for what is coming.*" Such rapid transitions between drought, snow, rain, and wind can
overstress infrastructure and increase the risk of cascading failures.
In the Danube Region, hazard interactions are complex and challenging to manage. For
contextual clarity, the interviews in the Danube Region included stakeholders based in Austria,
Hungary and Romania, representing basin-wide river authorities, agricultural and water
management agencies, NGOs, and academic institutions engaged in disaster risk
management. Floods frequently occur simultaneously across multiple catchments, often
overwhelming emergency response capacities. As one interviewee described, "*You have*
*multiple hazards occurring at the same time, meaning flooding from different sources. This*
*was a huge problem we observed in 2005, when not just one catchment flooded, but 30*
*catchments simultaneously. This created major challenges for disaster and emergency*
*management because there were insufficient human resources to respond.*" Droughts
followed by heavy rainfall reduce soil absorption and increase flood and erosion risks, creating
successive but closely linked hazards. Stakeholders also noted that the impacts of drought
are closely linked to water management and governance structures, illustrating that drought
risk is shaped as much by institutional decision-making as by environmental conditions (see
also Vargas & Paneque, 2019). Additionally, floods affecting industrial or contaminated sites
pose pollution risks for downstream communities. Stakeholders emphasized that such
combinations are becoming more frequent, but coordinated response is hindered by
fragmented risk management responsibilities across borders and institutional levels. Although
overarching guidelines exist for the Danube River Basin, implementation remains the
responsibility of individual states, resulting in differing regulatory frameworks, resource
allocation practices, and operational procedures. Within countries, responsibilities are
distributed across multiple agencies rather than a single authority, producing a complex and
fragmented DRM landscape that limits cohesive planning and response to cascading events.
In the North Sea, storms and high waves pose growing risks for offshore wind farms, maritime
operations, and port infrastructure. As offshore installations become denser, collision risks and
operational challenges during poor weather increase. Adverse conditions can delay
maintenance or emergency response, compounding the effects of power outages. Sea-level
rise and storm-driven flooding also threaten major ports, amplifying disruption across energy
and transport systems. These reflections were primarily raised by offshore energy operators
and maritime authorities, who stressed the operational implications of storm and wave
conditions for infrastructure continuity and safety.

The Canary Islands are also exposed to complex hazard interactions, including heatwaves that frequently coincide with Saharan dust haze events, which severely affect air quality, public health, and transportation. These events not only pose direct health risks but also drive up energy demand, increase the risk of wildfires, and intensify water scarcity, placing additional stress on critical infrastructure such as electricity networks and water systems. Volcanic activity introduces further cascading hazards, including lava flows, seismic activity, and toxic gas emissions, which disrupt communities, agriculture, and infrastructure resilience. In post-eruption landscapes, ash deposits in ravines and water networks significantly increase flood risks during heavy rainfall, heightening the potential for lahars. The insights presented here draw on interviews with civil protection authorities, agricultural cooperatives, tourism representatives, and energy and water operators, each highlighting different priorities shaped by their sectoral responsibilities.

While hazard combinations are unique for each pilot region, several recurring patterns emerge: floods often coincide with landslides in mountainous areas; heatwaves commonly occur alongside droughts, amplifying risks to agriculture, health, and energy systems; and coastal regions face intensified storm surges when combined with high tides or sea-level rise. These interconnected risks are further illustrated in storylines, such as those developed for the Veneto region (Casartelli et al., 2025).

3.2 Vulnerability characteristics

Vulnerability is shaped by the interplay of physical, social, economic, and environmental factors or processes (Box 1). These dimensions vary across pilot regions and sectors and are influenced by infrastructure conditions, demographic patterns, land use, governance structures, and the health of ecosystems. Together, they determine how exposed populations and systems are to hazards, how well they can respond, and how likely they are to recover. A summary of key interview-based insights across these four vulnerability dimensions is provided in Table 2. While Table 2 groups vulnerability into physical, social, economic, and environmental dimensions for analytical clarity, interviewees frequently described conditions that cut across and intertwine these categories. Many examples revealed how vulnerabilities reinforce each other through feedback processes: for instance, limited financial capacity (economic vulnerability) can prevent households or local authorities from maintaining or upgrading infrastructure, thereby increasing physical vulnerability, while fragmented governance or uneven access to services (often considered social or institutional factors) further amplifies both economic and physical risks. These interactions highlight that vulnerability operates as a dynamic and relational process, rather than as discrete categories, supporting the understanding of vulnerability as interconnected and systemic rather than isolated components.

Several interviewees pointed to weaknesses in infrastructure systems that reduce the ability to cope with hazards. In Veneto and the Danube Region, poorly maintained drainage and flood protection systems, combined with impermeable surfaces, were said to increase susceptibility to urban flooding. As one interviewee from the Danube Region explained, *"We have examples where the technical mitigation measures, such as levees, are maintained but there is no clear plan for their renewal or adaptation over time. This increases the physical vulnerability."* In Scandinavia, energy and transport infrastructure are vulnerable to sudden shifts between freezing and thawing conditions, especially in remote areas, due to less frequent maintenance and monitoring.

Conditions that reduce people's ability to avoid harm or recover quickly were also widely discussed. In several regions, interviewees highlighted how elderly populations, especially in rural or remote areas, are less able to respond during extreme events such as floods or heatwaves. In the Danube Region, social inequalities were linked to increased vulnerability: some communities lack early warning, access to services, or adequate housing. Another

interviewee from the Danube Region noted institutional limitations: *"The drought is already*
*announced when it exists... local authorities will only know when the problem exists, not*
*before."* These types of systemic and structural limitations make it harder for vulnerable groups
to prepare for, and recover from, hazard impacts.
Interviewees across all pilot regions highlighted how economic structures and dependencies
can amplify vulnerability to disruption. Agriculture, in particular, emerged as a sector with acute
sensitivity to increasingly unpredictable rainfall, droughts, and heatwaves. In the Canary
Islands, farmers and cooperative managers stressed the urgent need for accessible, context-
specific scientific knowledge to navigate these growing multi-hazard and climate challenges.
Similarly, in both Veneto and the Canary Islands, tourism (despite its economic importance)
shows structural weaknesses in coping with repeated shocks from floods, volcanic activity,
and extreme weather events. In Scandinavia, hydropower operators struggle to adapt to
shifting snowmelt patterns and water availability, while offshore energy systems in the North
Sea are grappling with rising maintenance demands under changing climatic conditions. As
one interviewee from the Canary Islands put it: "*Our socio-economic structure is so dependent*
*on sectors that are not only very sensitive to these threats, such as tourism and agriculture,*
*but also exacerbate the problem the way they are currently functioning.*" This sentiment
reflects a broader, recurring concern: high economic reliance on climate- and hazard-sensitive
sectors not only increases exposure but also reinforces unsustainable practices, such as land
and water overuse and infrastructure strain, which in turn heighten environmental stress,
erode adaptive capacity, and lock these regions into cycles of escalating systemic risk.
Ecosystem degradation and mismanagement were also cited as contributing to vulnerability.
In the Danube Region, degraded soils and reduced vegetative cover were said to intensify
runoff and erosion, undermining the landscape's ability to buffer heavy rainfall. One
interviewee explained, *"In the absence of vegetation cover, rainfall channels more easily into*
*the runoff, increasing flood risk. It's important to restore the different vegetation layers in order*
*to achieve better rainfall absorption."* These environmental conditions reduce the ability of
natural systems to moderate hazard impacts and recover after disturbances.
*Table 2: Example interview-based insights on vulnerability characteristics across the pilot regions. The*
*categories (physical, social, economic, environmental) are intended as analytical signposts related to*
*the definition of vulnerability as described in Box 1 rather than fixed boundaries; in practice,*
*vulnerabilities overlap and co-evolve.*

| Pilot Region | Physical Vulnerability | Social Vulnerability | Economic Vulnerability | Environmental Vulnerability |
|---|---|---|---|---|
| Veneto | Impermeable surfaces; aging water infrastructure; inadequate urban drainage | Aging population; limited preparedness and access to services in rural areas | Low financial resilience in agriculture and tourism sectors | Altered sediment flows; reduced ecological retention capacity |
| Scandinavia | Seasonal strain on energy infrastructure; limited resilience in cold-climate systems | Limited emergency access in remote areas; heating cost burdens | Hydropower sensitivity to climate variability; lack of diversification | Disrupted snowmelt patterns; reduced catchment stability |
| Danube Region | Poorly maintained flood risk reduction systems; compacted/degraded soils | Institutional delays; lack of targeted early warning for vulnerable groups | Institutional gaps in agriculture and industry | Reduced infiltration and flood buffering capacity |
| North Sea | Offshore infrastructure maintenance vulnerability; storm-sensitive energy | Operational pressure on maritime crews; gaps in contingency planning | Dependence on offshore energy and shipping; high cost of disruption | Sea-level rise undermining natural coastal protections |

| | systems; increasing risk of ship collisions due to higher offshore infrastructure | | | |
|---|---|---|---|---|
| Canary Islands | Isolated, non-redundant infrastructure; low shock absorption capacity | Energy poverty; informal housing; limited emergency access; fragmented governance | High dependence on tourism and export-oriented agriculture; growing stress from energy-intensive desalination and water scarcity | Deforestation; post-eruption ash accumulation; reduced soil moisture retention and land degradation |

Despite regional differences, common patterns emerge: infrastructure is strained by compounding hazards, vulnerable groups face barriers to coping with multi-hazard events, key sectors are sensitive to interacting risks, and ecosystem decline weakens natural buffering capacity. Several interviewees also pointed to resource constraints and institutional limitations as factors that shape vulnerability, suggesting that the ability to cope with risk is closely linked to access to funding, services, and governance capacity.

3.3 Changes in vulnerability and exposure

Interviewees across the pilot regions reflected on how vulnerability and exposure have shifted over time, shaped by changes in land use, institutional arrangements, socio-economic conditions, and cascading hazard events. Exposure has generally increased due to the spatial expansion of urban and economic activities into areas at risk of flooding, coastal storms, or volcanic activity. Vulnerability, by contrast, has evolved in more complex and context-specific ways - driven by structural inequalities, infrastructure aging, resource dependencies, and the erosion or recovery of local capacities. These findings resonate with the dynamic vulnerability typology of de Ruiter and van Loon (2022), which distinguishes three key processes that shape vulnerability over time: (1) underlying dynamics, (2) changes during long-lasting disasters, and (3) changes due to compound or consecutive disasters.

3.3.1 Underlying dynamics of vulnerability

The underlying dynamics of vulnerability refer to long-term structural shifts in socio-technical systems, governance, or ecosystems that gradually alter the ability of people and institutions to prepare for, respond to, or recover from hazard events (de Ruiter and van Loon, 2022). Across all pilot regions, interviewees pointed to increasing stress on infrastructure systems and gaps in institutional coordination.

In Veneto, urban expansion into marginal or low-lying areas has increased exposure to flood hazards, while aging infrastructure and limited maintenance investments reduce resilience. As one interviewee observed, "*We have areas that are progressively becoming more impermeable, mainly benefiting productive sectors. In fact, a wider part of the soil consumption occurs either for infrastructure or for productive settlements, which, however causes an increase of the impermeabilization of the territory and consequently of flooding phenomena due to the inability to retain water, which flows away, creating damage.*" This ongoing process of land consumption and loss of natural drainage capacity illustrates how long-term socio-technical and land-use changes systematically increase physical vulnerability to flooding over time.

Similarly, in the Danube Region, flood defenses face challenges due to delayed maintenance and funding constraints. As one interviewee noted, "*We do not sort that problem immediately, we always have to wait for something like some money to come out or some project to be developed. Then some other flood hit and then we have an even more devastating situation.*"

Governance fragmentation emerged as a key issue, particularly in the Canary Islands, where overlapping institutional responsibilities and limited strategic coordination were reported. One interviewee reflected: *"There are many administrations, some responsible for one thing, others for another, and in the end, you get lost"*. In the North Sea region, vulnerability stems not only from physical exposure but also from institutional and operational constraints. One interviewee highlighted the growing risks associated with rapid offshore energy expansion, noting, *"The main concerns are ship collisions - either between ships or with infrastructure."* This underscores how increasing infrastructure density challenges safe operations and heightens vulnerability.

### 3.3.2 Changes during long-lasting disasters

Long-duration events such as droughts, heatwaves, or volcanic eruptions gradually alter the vulnerability landscape by depleting resources, shifting dependencies, and reshaping livelihoods. The Canary Islands present a clear case: prolonged drought has driven a systemic shift toward desalination as a primary water source. While this reduces exposure to groundwater shortages, it introduces new vulnerabilities due to energy dependency. One stakeholder observed: "*Climate change is making us more vulnerable, but because we've changed sources*"; talking about the fact that the Canary Islands have moved from groundwater to desalination, which makes them more dependent on energy. This represents an adaptation-induced shift in vulnerability, where mitigation of one threat increases exposure to another.

A different dynamic was reported from the Danube Region, where long-term drought has eroded rural livelihoods and increased socio-economic fragility. Agricultural areas experience chronic water stress, while heatwaves reduce productivity and heighten financial risk. This gradual erosion of resilience was described as a "systemic vulnerability", not caused by a specific adaptation decision, but by the cumulative impacts of ongoing climatic stress and institutional inaction.

In Veneto, interviewees described how prolonged dry periods have exposed the fragility of water infrastructure and underscored the consequences of underinvestment. One stakeholder emphasized: *"There are hydric supplies, but the problem is that when there is a lot of water available (like heavy rain), we let most of the water leave very quickly. A proficient water management, without waste, is needed. The water network spills water from all sides; we've been saying this for decades, but we do very little."* This highlights how vulnerability accumulates over time due to inefficiencies in water storage and distribution, reinforcing the risks posed by long-term drought.

In La Palma (Canary Islands), the aftermath of the 2021 volcanic eruption illustrates how long-lasting disasters can shift vulnerability over time. Ash deposits and land loss not only affected physical infrastructure but also reduced long-term agricultural potential, especially for smallholders. Although the eruption was framed by some stakeholders as an opportunity to rethink land use and sectoral organization, interviewees highlighted that recovery efforts were fragmented and uneven, leaving many local residents without the institutional, financial, or technical support needed to adapt effectively. This reinforces pre-existing inequalities and new forms of socio-economic and territorial vulnerability.

### 3.3.3 Changes due to consecutive or compounding disasters

Consecutive or compound hazard events can fundamentally shift vulnerability. Consecutive events occur in sequence, such as a drought followed by flooding, while compound events happen simultaneously or share a common driver, like heatwaves and wildfires triggered by prolonged high temperatures. These shifts may emerge from cumulative impacts, cascading system failures, or reductions in coping capacity (de Ruiter and van Loon, 2022). As

mentioned above, in La Palma (Canary Island), volcanic ash from the 2021 eruption clogged ravines and drainage systems. Subsequent storms dramatically increased flood risk because the ash deposits obstructed the natural channels that normally carry runoff, forcing water to seek alternative paths that may flow toward populated areas. This situation illustrates how one disaster can increase vulnerability to the next.

The energy–water nexus was frequently mentioned across pilot regions. Power outages during heatwaves compromise water access, while high energy demands for desalination increase stress on grid systems. One respondent noted: "*Threats that affect the energy sector directly affect water. That wasn't the case 30 years ago*". These interdependencies amplify risk under compounding conditions. A similar concern was raised in the Veneto region, where one stakeholder observed: "*I wonder if the repetitiveness of calamitous events may be a characteristic, in the sense that at the moment the community, the system and the administrations may be resilient but when a second event strikes perhaps not anymore. In addition to the repetitiveness of events, the overlapping of events obviously makes (the region) much more vulnerable.*" This kind of hazard sequencing can prevent recovery between shocks, reinforcing a downward vulnerability spiral.

The interviews strongly support the typology proposed by de Ruiter & van Loon (2022), demonstrating that vulnerability across regions is not static, but shaped by the interplay of underlying, long-duration, and compounding processes. Across the pilot regions, common drivers include aging infrastructure, fragmented governance, sectoral interdependencies, and shifting climatic baselines. Importantly, efforts for disaster risk management and climate change adaptation - such as shifting to renewable energy or desalinated water - may themselves introduce new vulnerabilities if not considering asynergies (Haer and de Ruiter, 2024).

3.4 (A)synergies of DRR measures

Across the pilot regions, interviewees identified several DRR measures that created either synergistic or adverse (asynergistic) effects - highlighting how risk reduction in one domain can influence vulnerability or exposure in others (Table 3). These effects can be both intended and unintended, emphasizing the need for integrated and forward-looking planning in multi-hazard risk contexts.

In Scandinavia, hydropower was praised for its dual role in energy generation and flood regulation (i.e., synergies). As one respondent noted, "*We also see that, for instance, the hydropower producers are regulating the floods. So they are really reducing the risk of flooding*". However, the expansion of renewable energy systems, especially wind and hydropower, has introduced operational challenges for grid stability, especially during peaks of weather variability or during cold spells. These challenges illustrate how energy transition strategies can introduce new vulnerabilities if not coordinated with infrastructure resilience while also being regarded as providing synergies.

The Danube Region pilot offers a good example of synergy through ecological restoration. Wetland rehabilitation projects have reduced flood peaks, improved groundwater recharge, and supported biodiversity (Nichersu et al., 2022). Yet, some flood protection efforts - particularly levee construction - have been criticized for displacing floodwater downstream and reducing natural floodplain retention, thus intensifying drought conditions in certain areas. Stakeholders also noted that structural measures, such as levees, can sometimes create a false sense of security if they lead communities or decision-makers to underestimate residual risk, which can be counterproductive in the long term.

In the North Sea, offshore wind farms were highlighted as a major contributor to energy diversification and decarbonization. Yet the rapid expansion of this infrastructure raises spatial

and logistical concerns, particularly for shipping, environmental conservation, and emergency
access. One stakeholder explained, "*Space is increasingly contested. The expansion of*
*renewable energy is crucial, but its scale requires constant reassessment to avoid crossing*
*ecological thresholds*" - highlighting both synergies and asynergies are present.
In Veneto, the MOSE[1] barrier system has substantially reduced exposure to storm surges and
high tides in Venice. Stakeholders noted its positive effects on safeguarding infrastructure and
the cultural heritage of the city. However, concerns were also raised about ecological side
effects. As one interviewee explained, *"MOSE protects against high water (in Venice), but at*
*the same time prevents the (water) exchange at the mouths of the harbor between lagoon*
*waters and those of the open sea. This causes changes in the conformation of the lagoon*
*ecosystems."* This example illustrates how DRR measures can generate synergies (in this
case, protecting both people and cultural assets) while also producing asynergies (negative
impacts on the ecosystem).
In the Canary Islands, multiple synergies were identified. Strategic reforestation, when
implemented with native species and attention to landscape functions, was credited with
lowering wildfire risks and improving water retention. "*If we naturalize ravines, we avoid flood*
*risks, reduce fire risks, and improve biodiversity*" explained one interviewee. Reforestation was
also seen to support ecosystem resilience and reduce soil erosion, particularly in post-volcanic
landscapes. However, it was noted that poor management of reforestation - such as the use
of fire-prone non-native species - can reverse intended benefits. Meanwhile, reliance on
energy-intensive desalination to address water scarcity introduces new vulnerabilities related
to energy demand and supply.
Interviewees emphasized the importance of participatory planning and cross-sectoral
integration to maximize the synergies of DRR measures and prevent maladaptation.
*Table 3: Example interview-based insights on synergies and asynergies in Disaster Risk Reduction*
*Measures across pilot regions.*

| Pilot Region | Synergy | Consequence | Asynergy | Consequence |
|---|---|---|---|---|
| Veneto | MOSE system | Reduces storm surges, supports tourism and cultural heritage | MOSE system | Coastal erosion, disrupted sediment flows |
| Scandinavia | Hydropower generation | Supports energy security while regulating water levels for flood management | Renewable energy expansion (hydropower, offshore wind) | Grid instability, variability issues |
| Danube | Wetland restoration | Mitigates floods and droughts while enhancing biodiversity | (Urban) levees | Downstream drought intensification |
| North Sea | Offshore wind farms | Stabilizes the energy grid and reduces reliance on fossil fuels | Offshore energy congestion | Navigation risks, maintenance challenges |
| Canary Islands | Reforestation (native, strategic) | Reduces wildfire risks and protects ecosystems | Desalination plants | Energy strain during heatwaves |

These findings underscore that DRR measures rarely operate in isolation. Their synergies and
asynergies are shaped by broader systemic interdependencies and trade-offs. While
measures that integrate ecological and infrastructural functions tend to offer more consistent
co-benefits, others generate unintended trade-offs, particularly when systemic
interdependencies are overlooked. To synthesize these insights across the five pilot regions,
Table 3 presents a comparative overview of reported synergies and asynergies in DRR
interventions. It highlights how similar strategies can produce different outcomes depending

---

[1] The MOSE barrier system is a set of mobile gates that can be raised temporarily to protect the Venice Lagoon from high tides

on context, ranging from geographical setting to governance arrangements and sectoral
priorities.
This cross-case synthesis offers a foundation for the discussion that follows, where we reflect
on patterns, tensions, and opportunities in risk reduction practice across scales.
**4. Discussion**
The results highlight how multi-hazard risk is experienced and managed differently across pilot
regions and sectors. Interviewees described diverse hazard interactions, shifting
vulnerabilities, and the impacts of DRR measures. While some challenges were common, their
specific manifestations were highly context dependent. Although the number of interviewees
per pilot is limited, the qualitative insights offer early and context-rich signals of emerging risk
dynamics and governance challenges that are difficult to capture through quantitative
assessments alone. Rather than a singular risk landscape, the findings point to a complex,
evolving set of regionally embedded dynamics. This discussion reflects on four cross-cutting
issues that emerged from the interviews: persistent terminology challenges, divergent
stakeholder priorities, the dynamic nature of vulnerability and exposure, and the synergies and
asynergies of DRR measures. Together, these themes illustrate why integrated and adaptive
approaches are needed to address the realities of multi-hazard risk. Additionally, some
limitations of the research and its methodologies will be discussed.
4.1 Terminology challenges
A key challenge identified in this study is the inconsistent interpretation of core disaster risk
concepts - despite the use of standard definitions during the interviews. Across the pilot
regions, stakeholders used terms such as vulnerability, exposure, and resilience in diverging
ways, potentially shaped by their sectoral roles, institutional cultures, and past experiences.
As noted in the results (Section 3.2), some stakeholders conflated vulnerability with physical
exposure, while others framed it as a function of socio-economic status, institutional capacity,
or systemic dependency. For example, one interviewee from the North Sea region
emphasized vulnerability primarily in relation to physical exposure to storms, illustrating how
conceptual interpretations can vary between a focus on hazard impacts and broader systemic
factors.
Similar inconsistencies were observed in how stakeholders described multi-hazard
interactions. While some emphasized acute cascading events (e.g., flash floods triggering
pollution events), others focused on longer-term, compounding processes, such as heat and
drought undermining energy and water security. These divergent framings reflect the
complexity of risk perception across regions and sectors - and underscore the challenges of
aligning terminology in practice.
As noted by Staupe-Delgado (2019), revisions to the UNDRR terminology have introduced
greater clarity in some areas, while removing or reshaping long-standing concepts in others.
Our findings confirm that despite these formal efforts, inconsistencies persist in practice, often
due to discipline-specific language, institutional memory, or operational needs. Without a
shared conceptual framework, coordination between sectors and regions becomes more
difficult, and well-intentioned risk reduction efforts may diverge or even conflict (Kelman,
664 2018).

This ongoing challenge highlights the need for more than just standardized definitions; active
engagement with diverse stakeholder groups is crucial to fostering conceptual clarity in multi-
hazard risk understanding and governance. Our findings support existing calls (Kelman, 2018;
Šakić Trogrlić et al., 2024) for developing not only shared conceptual frameworks but also
mechanisms for ongoing, participatory dialogue (Bharwani et al. 2024)
4.2 Stakeholder priorities and context-specific perspectives
Across the five pilot regions, the interviews revealed that multi-hazard risk is interpreted and
prioritized in diverse ways, shaped by context-specific challenges, institutional roles, and lived
experiences. Interviewees tended to emphasize the hazards, systems, or sectors most familiar
to them. For example, in Scandinavia and the North Sea, energy sector representatives
focused on the resilience of hydropower and offshore infrastructure, often framing vulnerability
in terms of operational continuity and supply chain stability. While these emphases were not
always articulated as formal priorities, they emerged clearly through the themes and examples
stakeholders chose to elaborate on during interviews.
These differences highlight the need for context-specific approaches to multi-hazard risk
governance. However, as Santos et al. (2024) observe, integrating diverse sectoral and
regional perspectives remains a persistent challenge - often leading to fragmented strategies
or overly generalized recommendations. Our findings support this: while stakeholders clearly
identified key challenges in their regions, the lack of shared platforms and mandates often
limited their ability to act on them. Past disaster experiences also shaped stakeholder
perspectives. For example, in the Canary Islands, the 2021 La Palma eruption highlighted
long-term vulnerabilities in recovery and land use conflicts.
As noted by Šakić Trogrlić et al. (2024), competing institutional mandates and resource
constraints frequently hinder the translation of such lessons into more coordinated action.
Interviewees pointed to fragmented governance, limited funding, and a lack of sustained
cross-sector collaboration as key barriers. While structured dialogue remains essential, our
findings suggest that deeper institutional reform might be needed. Shared mandates, financing
mechanisms, and durable spaces for coordination are critical for building adaptive, integrated
approaches to multi-hazard risk governance (Elkady et al., 2024).
4.3 Context-specific dynamics of hazard interactions, vulnerability and exposure
The interviews underscore that vulnerability and exposure are deeply context-specific and
shaped by regional development patterns, institutional arrangements, and sectoral
dependencies. These local dynamics interact with distinct hazard profiles, reinforcing that risk
cannot be assessed or managed without close attention to context (Thompson et al., 2025).
Hazard interactions varied significantly across pilot regions. The Canary Islands face
concurrent volcanic activity, heatwaves, and haze; Scandinavia contends with rapid freeze–
thaw cycles; and the North Sea region experiences intensifying storms. These hazard
patterns, shaped by climatic and non-climatic processes, underscore that hazards
themselves, not just exposure or vulnerability, are regionally specific (Gill et al., 2020).
Vulnerability, too, evolves in response to dynamic conditions. In the Canary Islands, reliance
on desalination has reduced groundwater dependence but introduced new energy-related
fragilities. Elsewhere, aging infrastructure and institutional fragmentation constrained
adaptation.
These regionally specific insights reaffirm that vulnerability and exposure are not static, but
evolve through structural, institutional, and environmental changes. Our findings therefore
support a long-standing body of research that understands vulnerability as socially and
politically constructed (e.g., Kelman, 2018). Stakeholders repeatedly linked vulnerability to
funding priorities, regulatory decisions, and power imbalances that shape whose risks are
addressed and whose are overlooked. For instance, maintenance backlogs in the Danube
Region were described as a direct consequence of political choices rather than technical
limitations, while in the Canary Islands and Veneto, interviewees emphasized that
communities with limited economic resources face greater difficulty adapting or recovering.
Such examples illustrate how institutional and economic arrangements actively produce
unequal distributions of risk. Interviewees described how risk accumulates due to institutional
inertia, slow adaptation, and the unintended effects of risk reduction measures, patterns that
clearly align with the dynamic vulnerability typology of de Ruiter and van Loon (2022). Rather
than capturing these shifts through models alone, stakeholder narratives offer a crucial lens
into how slow-onset and systemic drivers of risk manifest in practice, particularly in contexts
where top-down assessments fall short.
4.4 Cross-regional synthesis: emerging systemic patterns
Across the five pilot regions, several recurring patterns emerged despite major contextual
differences, demonstrating shared systemic challenges in managing multi-hazard risk across
Europe. Hydro-meteorological hazard interactions (particularly flooding driven by rainfall,
snowmelt, or rapid freeze-thaw cycles) were frequently mentioned across Veneto,
Scandinavia, and the Danube Region. In these regions, climate variability and seasonal
extremes increasingly challenge infrastructure and emergency planning (Forzieri et al., 2018).
Coastal and island regions, such as the North Sea and Canary Islands, emphasized
compound hazards related to sea-level rise, storm surges, and heatwaves, often occurring
alongside cascading system impacts. These challenges mirror findings from broader climate
risk assessments, which highlight the growing exposure of coastal zones and critical
infrastructure to compound risks (Forzieri et al., 2018; Pal et al., 2023).
While climate-related hazards dominated stakeholder concerns across all pilots, non-climatic
risks (such as seismic and volcanic activity) remained central in specific contexts, especially
in the Canary Islands. The 2021 La Palma eruption, for example, exemplified how cascading
hazards (e.g., lava flows, ashfall, floods) strain emergency systems and reshape long-term
vulnerability. Infrastructure vulnerability also emerged as a shared concern. Across all pilots,
stakeholders cited inadequate maintenance, aging systems, and lack of redundancy as major
amplifiers of risk (Verschuur et al., 2024). Institutional fragmentation was likewise consistently
mentioned (particularly in the Danube Region and Canary Islands) where overlapping
mandates and limited coordination hindered effective risk management (Papathoma-Köhle et
al., 2021). These findings resonate with broader research on governance gaps in European
disaster response (e.g., Vollmer et al., 2024). Economic vulnerability was commonly linked to
sectoral dependencies: many pilots noted how key sectors like tourism, agriculture, and
energy are both highly exposed and interdependent with environmental conditions.
Stakeholders also emphasized that disaster risk reduction measures often create both
synergies and asynergies. Nature-based solutions (such as wetland restoration in the Danube
or strategic reforestation in the Canary Islands) were widely viewed as effective and context-
sensitive. In contrast, large-scale grey infrastructure, like the MOSE barrier in Venice or
offshore wind expansions in the North Sea, were praised for reducing targeted risks but also
critiqued for their ecological trade-offs and unintended consequences. These reflections
reinforce the understanding that multi-hazard risk is dynamic and systemic, shaped by the
intersections of physical hazards, institutional capacity, and socio-economic dependencies.
These cross-regional parallels illustrate that multi-hazard risk governance challenges are not
isolated to individual contexts but stem from wider systemic drivers that manifest across
sectors and scales. Recognizing these shared patterns provides a foundation for collective
learning across regions while acknowledging the importance of local specificity.
4.5 Navigating synergies and asynergies in Disaster Risk Reduction
We identified examples of both synergies and asynergies. Synergies were observed most
consistently in measures that integrate ecological and infrastructural functions. For example,
in the Danube Region, wetland restoration was reported to reduce flood peaks and support
groundwater recharge, while simultaneously enhancing biodiversity - an example of nature-

based DRR with cross-sectoral co-benefits. At the same time, the study revealed that DRR measures can generate asynergies if their systemic impacts are not fully considered. These include structural trade-offs, increased interdependencies, or negative spillover effects across sectors. For example, in the Veneto pilot, the MOSE flood barrier was shown effective in reducing tidal flooding in Venice, yet interviewees expressed concern over its disruption of sediment transport and lagoon ecology - impacts that may degrade long-term coastal resilience. Our findings reinforce growing concerns in the literature that DRR measures, when not approached systemically, may result in unintended consequences that compound multi-hazard risk (Ward et al., 2020; de Ruiter et al., 2020; Simpson et al., 2021).

These dynamics echo the concerns raised in the Midterm Review of the Sendai Framework (UNDRR, 2023), which highlights that despite progress in DRR implementation, risk emergence continues to outpace risk reduction. Sectoral silos, limited foresight, and short-term planning remain persistent barriers. Related literature on maladaptation similarly cautions that risk-reducing strategies - especially those addressing climate change - may result in rebounding vulnerability, the redistribution of risk, or adverse externalities for non-targeted groups (Schipper, 2020; Simpson et al., 2021).

To avoid such outcomes, DRR measures should move beyond single-hazard perspectives. Integrated approaches that assess both the synergies and asynergies of interventions across sectors, time, and space are critical (Cremen et al., 2023). As the findings of this analysis demonstrate, risk-informed decision-making should account not only for direct hazard impacts but also for second- and third-order consequences of policy choices. Without such foresight, even well-intentioned measures may undermine resilience and exacerbate multi-hazard risk.

4.6 Limitations

This study was designed as a structured yet flexible screening of stakeholder perspectives across five diverse pilot regions. While the methodology allowed for the capture of rich, context-specific insights, it also introduced limitations related to cross-regional consistency, stakeholder diversity, and the interpretive nature of qualitative data. Ensuring coherence across regions with different hazard profiles, institutional arrangements, and cultural contexts required a balance between a shared interview framework and local differences. Pilot leads played a key role in maintaining this balance, facilitating interviews, providing contextual interpretation, and supporting translation when interviews were conducted in local languages. These translations were collaboratively reviewed to ensure accuracy, but subtle differences in expression and emphasis may still have influenced interpretation.

Stakeholders were selected using purposive sampling to reflect sectoral diversity and relevance to disaster risk governance. However, participation was limited to those already engaged in the MYRIAD-EU project or accessible through pilot networks. As such, the findings represent a broad but not exhaustive range of perspectives. Group interviews, while valuable for dialogue, may have also introduced power dynamics that influenced individual contributions. Although a common analytical framework was used, regional variation in institutional language, sectoral priorities, and governance contexts complicated direct comparison. While terminology challenges are discussed in Section 4.1, from a methodological perspective these inconsistencies added complexity to thematic synthesis.

While this study aimed to ensure sectoral diversity, the number of interviewees per region was limited. In larger and more complex pilots such as the Danube Region, where risk contexts span multiple countries, the findings reflect only a narrow segment of perspectives. These constraints underscore the need for further region-specific studies that build on these exploratory insights.

Additionally, the analysis was primarily interpretive and illustrative, not intended to provide statistically generalizable findings across Europe or even within the individual pilot regions. Despite these limitations, the process generated valuable insights into stakeholder priorities, risk framings, and evolving vulnerabilities. It also highlighted the importance of trust, co-design, and methodological flexibility in qualitative disaster risk research - lessons that can inform future efforts in multi-hazard, multi-actor settings.

## 5. Conclusion

This study provides empirical insight into the evolving nature of multi-hazard risk by examining hazard interactions, changes in vulnerability and exposure, and the synergies and asynergies of DRR measures across five European pilot regions. The findings underscore that risk is not static but shaped by dynamic interlinkages between physical, social, economic, and institutional systems. These dynamics manifest differently across regions, depending on historical experiences, governance structures, and development trajectories (e.g., urbanization). Nonetheless, commonalities also emerged: aging infrastructure, fragmented institutions, climate-induced stressors, and reliance on exposed economic sectors increase vulnerability across all pilot regions.

A key contribution of this study lies in illustrating how long-term vulnerability dynamics - whether driven by gradual degradation, long-lasting disasters, or the cascading effects of compounding hazards - intersect with shifting exposure patterns and systemic dependencies. While some DRR measures provide synergies, others create unintended trade-offs that reinforce vulnerabilities elsewhere or over time. The interviews clearly show that DRR measures, if not carefully designed, can increase future risk - for example, desalination increasing energy dependency in the Canary Islands.

By foregrounding stakeholder insights, this study sheds light on the often-overlooked dynamics that shape multi-hazard risk across time and space. The results show that risk is rarely static: it evolves in response to deep-rooted structural factors, gradual environmental degradation, and the unintended consequences of adaptation measures. These findings validate the need to move beyond event-based assessments and incorporate dynamic vulnerability frameworks into DRR planning. Importantly, this requires acknowledging that vulnerability is shaped by political-economic decisions and social inequalities, rather than treating it as an inevitable or apolitical outcome. In particular, understanding how vulnerabilities emerge through institutional concerns, cascading hazards, and shifting socio-economic dependencies, such as reliance on desalination or offshore infrastructure, can inform more flexible, integrated, and participatory governance. Qualitative insights from those directly engaged in managing risk not only enrich the evidence base but also help anticipate future challenges that may otherwise remain invisible in quantitative models. Embedding such perspectives is essential not only for reducing disaster risk, but for enabling adaptive governance in the face of accelerating and interconnected crises.

While the results are inherently context-specific, the comparative perspective illustrates that shared systemic issues underpin multi-hazard risk across regions. The ability to identify transferable lessons, such as the need for cross-sector coordination, attention to cascading infrastructure dependencies, and recognition of socio-political drivers of vulnerability—highlights the value of stakeholder-based qualitative insights for advancing DRM/DRR strategies beyond local scales.

**Author Contribution**

NvM led the research design and the writing of the manuscript, with contributions from MdR, WJ, and PJW. MdR, WJ, SG, RC and PJW also contributed to the conceptual framing of the study. VC, SG, RC, ASD, DG, ALD, JDP, PDA, TFA, SGG, RHM, CAA, JJDH, NP, ST, SHS, LM, LMt, KR, TS, RŠT, ST, DRMdlL, and YHG conducted the stakeholder engagement and interviews and reviewed the transcripts for contextual accuracy. RC and SG coordinated the interview process and provided oversight across the pilot regions. NvM transcribed and translated the interviews and conducted the data analysis. All authors contributed to reviewing and editing the manuscript.

**Competing Interests**

At least one of the (co-)authors is a member of the editorial board of Natural Hazards and Earth System Sciences.

**Acknowledgements**

We gratefully acknowledge all interviewees for their valuable time and thoughtful contributions to this research.

**Financial Support**

This research has been supported by the Horizon 2020 project MYRIAD-EU (grant no. 101003276).

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

**Supplementary Material**

Interview Themes and Questions

Theme 1: Hazard Combinations
- What combinations of hazards are important in your region when considering disaster risk, in your organization, or in the context of your work?
- Why are these combinations of hazards important in your region?
- Are there certain combinations of hazards that are becoming more important?
- How are these hazard combinations considered when designing disaster risk management measures?

Theme 2: Vulnerability Characteristics
- In your opinion, what are the most important characteristics that determine vulnerability to different hazards in your region?
- How do these important vulnerability characteristics differ between different economic sectors?
- Do you have any examples from your region or cases in which vulnerability (of certain groups, people, economic sectors, etc.) turned out to be different than perceived beforehand?

Theme 3: Changes in Exposure and Vulnerability Characteristics
- Can you provide examples of situations in which vulnerability or exposure conditions changed in your region due to changes in underlying socioeconomic conditions (e.g., economic recession, land use change, conflicts, migration)?
- Can you provide examples of situations in which vulnerability or exposure conditions changed in your region during long-lasting disasters (e.g., heatwaves, droughts, COVID-19)?
- Can you provide examples of situations in which vulnerability or exposure conditions changed in your region as a result of combinations of hazards?

Theme 4: Synergies and Asynergies of Disaster Risk Reduction Measures
- Can you provide examples from your region of cases in which measures taken to reduce risk from one hazard also had beneficial effects for another hazard?
- Can you provide examples from your region of cases in which measures taken to reduce risk from one hazard also had negative effects for another hazard?