# Peer review of "Bridging Science and Practice on Multi-Hazard Risk Drivers: Stakeholder Insights from Five Pilot Studies in Europe"

_EGUsphere, 2025_

## Author Comment (AC1)

**Reviewer 2**

| # | Comment | Response | Changes | Page, line |
|---|---------|----------|---------|------------|
| 0 | Overall, this is a well-written, polished, clear and important piece drawing on stakeholder insights from various case-studies ("pilot regions") to discuss multi-risk contexts and their complexity in those regions. Given the need for both highly context-sensitive DRR and DRM processes on the one hand, and a generalizable approach towards better understanding and dealing with an increasingly complex disaster risk context, the article provides important input for both the academic study and the practical handling of multi-hazard risks. An aspect that the authors may further develop is how they introduce the various stakeholders and the respective risks they face. As they point out, risks are multi-faceted and complex, and affect people in different ways. However, this differentiation of who is affected by what kind of risk, why that risk matters and for whom, could be highlighted a bit further (see comments below for specifics). Accordingly, I recommend the article be published after some minor revisions. | Thank you very much for the positive assessment and these thoughtful suggestions. We are pleased that the importance of our work is recognized. We have taken on board the recommendation to further highlight which stakeholders face which risks and why those risks matter to them. In the revisions, we have more clearly differentiated the perspectives of various actors in the text (as detailed in the specific responses below). Overall, we agree that balancing context-specific detail with generalizable insights is crucial, and we have aimed to strengthen the manuscript on that front. | N/A (general comment addressed through specific changes below) | N/A |
| 1 | From line 244 (3. Results), onward: It would be interesting to understand better who is actually dealing with what kind of hazard/risk. When the authors mention the Danube region (line 274), which is arguably an immense space (compared to e.g., Veneto), it would be interesting to know where and how the hazard interactions are complex to manage, and for whom. This also matters, since | Thank you very much for this thoughtful suggestion. We agree that clarifying which stakeholders are associated with each hazard example enhances the interpretation of the results and avoids presenting hazards in isolation from those who experience or manage them. In response, we have revised Section 3 to explicitly identify the stakeholder | Added stakeholder contextualisation in regional narratives within Section 3.1 (Hazard combinations), specifying the institutional roles and sectoral perspectives linked to the described risks (e.g., civil | Section 3.1, Section 4.2 |

| | | | | |
|---|---|---|---|---|
| | NGOs, academics, city administrators, and local municipalities may all have different conceptions of what makes a risk, and why that risk matters. While this is most obvious in the Danube region (given its size), similar things can be said about the North Sea, and the Canary Islands. While it makes sense that the authors only focus on a specific risk for a specific number of stakeholders in one publication, it would be useful to contextualize those risks with the various actors involved, rather than leaving the hazard "speak for itself". | groups whose perspectives inform each regional example. This includes clarifying the institutional scope and actor roles in the Danube Region (e.g., basin-wide authorities, agricultural and water management agencies, NGOs and academic institutions), and adding comparable contextualization in the Veneto, Scandinavia, North Sea and Canary Islands pilots. These additions make clear for whom hazard interactions are complex to manage and why certain risks matter to different stakeholders. Additionally, we strengthened the Discussion by adding a short subsection (Section 4.2) that reflects on differences in risk prioritisation between stakeholder groups, reinforcing the importance of context-specific perspectives in multi-hazard governance. | protection authorities, infrastructure operators, offshore energy developers, agricultural and tourism sector representatives).

Added Discussion subsection 4.2 to synthesise how risk priorities differ across actors and regions. | |
| 2 | On a more general level (and granted, the authors somehow mention this in the discussion/limitations section of the article), it may be interesting to reflect a bit more on the generic value of assessing and comparing highly heterogeneous set of stakeholders and regions to one another. Not that it may not have epistemic value to do so – but it may be interesting to highlight a bit more, why the information garnered from these interviews is valuable for DRM/DRR development beyond the fact that it is complicated and context-sensitive (a fact, that the article expresses nicely). | Thank you for this insightful comment. We agree that it is important to more clearly articulate the added value of comparing heterogeneous contexts beyond illustrating complexity. In the revised manuscript, we strengthened the Discussion to explain how comparative stakeholder perspectives help identify shared systemic challenges (such as governance fragmentation, infrastructure vulnerability, and interdependencies across critical sectors) that transcend individual regions. We also highlight how qualitative stakeholder insights reveal institutional and socio-economic dynamics that may not be visible in quantitative analyses, thus offering transferable lessons for DRM/DRR. These | We streamlined and expanded Section 4.4 to synthesize cross-regional patterns and demonstrate their relevance for generalizable DRM/DRR insights. We also refined the Conclusion to emphasize the contribution of comparative qualitative analysis for multi-hazard risk governance across scales. | Section 4.4, Section 5 |

| | | additions clarify why cross-regional comparison is valuable for broader risk governance development, not only within individual contexts. | | |
|---|---|---|---|---|
| 3 | Should "in progress" be cited? (line 135) | We acknowledge this concern. The reference in question ("Ciurean et al., in progress") refers to a project manuscript in preparation. We have adjusted the text to clarify this status. In the revised manuscript we now label it as "Ciurean et al., in preparation" instead of "in progress," which is a more standard way to cite a work that is not yet published, we have also added the full working title. We will of course update the reference to a formal citation once that work is published, but for now it remains an in preparation citation included for completeness. | Revised the citation wording for the pending reference from "in progress" to "in preparation" to clearly indicate it's a work in preparation. Additionally, the full working title has been added. | Section 2.1 |
| 4 | Double period (line 456) | We have corrected this typographical error. The duplicate period in the manuscript has been removed so that the sentence now ends with a single period. | Deleted the extra "." and fixed the punctuation at the end of the sentence on multi-hazard management challenges. | Section 3.3.2 |

---

## Author Comment (AC2)

**Reviewer 1**

| # | Comment | Response | Changes | Section |
|---|---------|----------|---------|---------|
| 0 | Overall, the quality of the paper is good and the work is interesting (although not entirely novel, subject to academic trends). Some case studies received more attention than others, leading me to believe that maybe the paper could have benefitted from a more narrow focus on case studies (1-3), to allow space for more in-depth exploration of contextual issues. Nonetheless, it's refreshing to see a practical take on the issue of cascading/multi-hazard risks – a body of knowledge that seems to be populated by frameworks and concepts nobody uses. However, some edits would be required before publication. | Thank you for the positive overall assessment and constructive suggestions. We appreciate the feedback and have made a number of revisions in response to the specific points raised below (including clarifying the focus of case studies, as addressed under Comment 13). | N/A (general appreciation; specific issues are addressed in responses and revisions below) | N/A |
| 1 | Does the paper address relevant scientific questions within the scope of ESD? Yes | Thank you for confirming this. | N/A | N/A |
| 2 | Does the paper present novel concepts, ideas, tools and data? Not novel, but the perspective is interesting. | Thank you for the feedback. | N/A | N/A |
| 3 | Are substantial conclusions reached? Partly – the limitations of available data should be highlighted more clearly. I understand the emphasis on stakeholders' perspectives, but they provide a flimsy ground to make sweeping claims (see point 4). | Thank you for this valuable comment. We agree that the scope and limitations of our qualitative data needed to be more clearly articulated. In the revised manuscript, we now explicitly emphasize that the findings are illustrative rather than statistically generalizable, and we clarify that they depend on the scale and composition of the interviewed stakeholders. We also | Added a clarifying sentence in the Introduction noting that the study adopts an exploratory qualitative approach and is not intended to be statistically generalizable. | Section 1, Section 4 (Opening), Section 4.6 |

| | | strengthened the rationale for integrating stakeholder perspectives despite the limited sample size, explaining their value for identifying emerging dynamics that are not yet visible in quantitative datasets. Finally, we expanded the Limitations section to explicitly acknowledge constraints such as sample size, stakeholder diversity, contextual depth, and translation considerations. | Strengthened the Discussion to articulate the value of qualitative insights for revealing risk dynamics not captured in quantitative models.

Expanded the Limitations section to clearly outline dataset constraints and emphasize that results are indicative rather than conclusive. | |
|---|---|---|---|---|
| 4 | Are the scientific methods and assumptions valid and clearly outlined? Partly. The issue of terminology is acknowledged vastly too late (section 4.1), and the approach to handing stakeholders a glossary clearly turned out to be counterproductive. The author's initial approach risked reinforcing gaps between expert discourse and practice, rather than exploring and assessing differences in interpretation. To make scientific knowledge useful, aligning scientific and local knowledge should be a priority. This should be highlighted earlier in the paper.

Secondly, the readers would benefit from understanding what countries were involved in regions described. Considering the geographic, political, and socio-economic diversity of the Danube Region in particular, it is difficult to interpret interview responses in the absence of contexts (especially when the authors make claims regarding "fragmented risk management systems which limit | Thank you for this helpful comment. In the revised manuscript, we moved the reflection on terminology earlier into the Methods section to clarify how definitions were used and interpreted in interviews. We also specified the stakeholder representation within the Danube Region and expanded the Results section to explain what is meant by fragmented risk management systems, why fragmentation occurs, and how it affects coordinated response, based on stakeholder testimony. | New paragraph on terminology added to Section 2.2

Added clarification of stakeholder countries and institutional types in Danube paragraph in Section 3.1

Expanded explanation of DRM fragmentation in Section 3.1 | Section 2.2, Section 3.1 Section 3.1. |

| | | | | |
|---|---|---|---|---|
| | effective responses"). Fragmented systems where? Fragmented how/why? Evidenced by what/whom? | | | |
| 5 | Are the results sufficient to support interpretations and conclusions? Partly, but more in-depth discussion is required. For example, the missing references to socio-economic conditions in the context of vulnerability is surprising – particularly when the data suggests this specifically (e.g., Danube and the "maintenance funding constraints" underpinning infrastructural vulnerability). These are not some random system changes, but rather reflect politico-economic decision making that (usually for the benefit of the free markets) seeks to cut public funding for services. This applies to social vulnerability as well: people do not choose to live on unsafe land, in poor quality housing, or choose not to build back better. They do so because they tend to have limited financial means (by-product of economic arrangements that are, again, not without intention). It is as if the authors go out of their way to avoid mentioning economic issues, rather choosing to point to "systemic transformations" which leave the reader to decide what the issue is. However, this is an academic article, not a Rorschach's test. | Thank you for this important comment. We agree that the role of socio-economic conditions in shaping vulnerability required clearer articulation. To maintain analytical integrity, we did not insert interpretive political-economic analysis directly into the Results section, as this would have gone beyond what the interview data support. However, we clarified in the Results that several stakeholders pointed to resource constraints and institutional limitations as drivers of vulnerability. We then substantially strengthened the Discussion (Section 4.3) to explicitly address the political-economic production of vulnerability and linked this analysis to relevant literature (e.g., Wisner, 2016; Kelman, 2018). A brief addition was also made in the Conclusion to highlight the implications of these dynamics for equitable DRR. | Added a short clarifying sentence in Section 3.2 (Vulnerability characteristics) noting that stakeholders highlighted resource and capacity constraints influencing vulnerability.

Expanded Section 4.3 (Discussion) to explicitly discuss socio-economic and political-economic determinants of vulnerability, supported by literature.

Added a brief statement in the Conclusion referencing the implications of unequal resource distribution for DRR pathways. | Section 3.2, Section 4.3, Section 5. |
| 6 | Is the description of experiments and calculations sufficiently complete and precise to allow their reproduction? Yes (although the paper would benefit from including the codes used as an annex). | Thank you for this helpful suggestion. We have expanded the description of our coding procedure in the Methods section to make the analytical steps more transparent and reproducible. Given the qualitative nature of the study and ethical considerations surrounding interview data, we chose not to include raw code files as supplementary | Added a detailed description of the two-stage deductive–inductive coding procedure in Section 2.3 (Data Analysis), including how codes were developed, refined, and reviewed. | Section 2.3 |

| | | material, but the revised text clarifies how the coding was conducted and reviewed. | | |
|---|---|---|---|---|
| 7 | Do the authors give credit etc. Yes | Thank you! | N/A | N/A |
| 8 | Does the title clearly reflect the contents? Yes, although the bridging appears to come in as an afterthought (the issue of section 4.1). | We believe this concern has been addressed through the revisions described above. By moving the discussion of terminology and the need to bridge scientific and local knowledge earlier (see our changes for Comment 4), we make the "bridging" aspect more integral to the narrative rather than an afterthought. These changes clarify from the Methods onward how we connect science and practice, which should reinforce the title's relevance. | No specific additional change beyond those made for Comment 4. (The earlier introduction of the terminology/gap discussion in Methods now ensures the bridging theme is evident.) | N/A |
| 9 | Concise and complete summary? Yes. | Thank you! | N/A | N/A |
| 10 | Overall presentation: well structured and clear? Yes | Thank you! | N/A | N/A |
| 11 | Is the language fluent and precise? One minor issue – the more commonly acknowledged term for "long/fast duration events" is slow/fast onset event/disaster. Also in the beginning of the paper, it perhaps would be best to start with risks are, rather than "risk is". | Thank you for this helpful comment. We agree that the opening sentence reads more clearly in the plural form and have therefore changed "risk is increasing globally" to "risks are increasing globally."

Regarding the terminology for temporal characteristics of disasters, we chose to retain the phrasing "long-lasting" and "rapid-onset" disasters rather than adopting "slow-onset / fast-onset," as these terms align with the conceptual framing of de Ruiter & van Loon (2022), whose framework underpins our | Adjusted the wording in the first lines of the Introduction from "risk is increasing globally" to "risks are increasing globally" | Section 1 |

| | | | |
|---|---|---|---|
| | analysis. Maintaining consistency with this terminology ensures coherence across the manuscript and avoids conceptual ambiguity between event onset and duration. We clarified this framing in the related section to support readability. | | |
| 12 | Are mathematical formulae, symbols, abbreviations, and units correctly defined and used? N/A, however the usage of vulnerability concept remains rather inconsistent. Firstly it notes vulnerability as a process, and the goes onto box 1 to provide a neat categorisation for the purposes of discussion. Yet, social vulnerability (for example) is not easy to separate from economic vulnerability, and economic vulnerability tends to produce physical vulnerability (e.g. quality of housing). Whilst tidy for analysis, the point of the paper is to emphasise how messy and interconnected these issues are. The authors should revisit the results section with this in mind (without forgetting the issue of economy!). | Thank you for highlighting the need to more clearly communicate the interconnected nature of vulnerability dimensions. We agree that vulnerability is best understood as a dynamic and relational process, and that analytical categories should not imply neat separation. In response, we have updated Section 3.2 of the Results to explicitly discuss the overlaps and feedbacks between social, economic, physical, and environmental vulnerabilities. This includes an example of how economic precarity can drive physical vulnerability (e.g., inability to maintain infrastructure or secure safe housing), and how social/institutional factors can amplify these processes. We also clarified in Box 1 that the categories are used for analytical purposes only. These revisions ensure that the paper reflects the messy, interconnected nature of vulnerability that stakeholders described and avoids underrepresenting economic drivers. | Added paragraph in Section 3.2 (Vulnerability characteristics) discussing vulnerability interconnections and feedbacks

Clarified intent of Box 1 terminology in Table 2 header | Section 3.2 |
| 13 | Should any part of the paper be clarified, reduced, combined, eliminated? See above. Also, the inclusion of all case studies has been done for the sake of the project, but for the sake of the paper and detail, focusing on 1-3 may be more beneficial (if feasible). | Thank you for this suggestion. We appreciate the concern that including all five pilot regions may risk overwhelming the reader with detail. However, the purpose of this paper is to identify cross-regional patterns and systemic dynamics in multi-hazard risk across diverse | Streamlined descriptive content in the Results to improve focus and reduce repetition | Section 4.4, Section 3 |

| | | European contexts. Reducing the analysis to only 1–3 regions would limit our ability to identify commonalities and differences across contexts, and would weaken the comparative insights that form the core contribution of the paper and of the MYRIAD-EU project.

To address the underlying concern about clarity and focus, we have streamlined the Results section to minimize repetition and strengthened the synthesis across regions. We have added an explicit cross-regional synthesis subsection in the Discussion (Section 4.4), which draws together shared patterns, governance challenges, and lessons from across all five pilots. This helps guide the reader to the overarching insights without requiring case-level detail to be revisited repeatedly. We believe these revisions maintain readability and analytical focus while retaining the breadth necessary to support the paper's comparative contribution. | Added explicit cross-regional synthesis subsection in the Discussion (new Section 4.4, "Cross-regional synthesis: emerging systemic patterns")

Strengthened thematic framing to clearly highlight shared insights across pilots | |
| 14 | Are the number of quality references appropriate? Yes. | Thank you! | N/A | N/A |
| 15 | Is the amount and quality of supplementary material appropriate? Would benefit from the inclusion of codes | Please see our response to Comment 6 above. We have incorporated an explanation of the coding process used in the study (in the Methods section) instead of providing the raw code list as supplementary material. This should address the intent of the comment by giving transparency into how the qualitative data were handled. | As noted for Comment 6, we added a description of the coding methodology in the main text (Methods section) to document how themes were derived and handled. | N/A |

| | | | |
|---|---|---|---|
| 16 | Other minor thoughts: p7, droughts are generally a water management problem as much as they are an environmental one https://www.mdpi.com/2071-1050/11/2/308. Does not hurt to mention this. | Thank you for this observation. We agree that drought risk is shaped as much by water management and governance decisions as by environmental processes. In response, we added a brief clarification in Section 3.1 acknowledging the role of water governance in shaping drought impacts and referencing Vargas & Paneque (2019). | Added a sentence in Section 3.1 noting that stakeholders highlighted the importance of water governance and institutional decision-making in drought-related risk. | Section 3.1 |
| 17 | P12 and discussions regarding levees: infrastructural measures themselves can create a false sense of security and therefore becoming counterproductive to DRR . Could be an interesting issue to explore. | Thank you for this thoughtful suggestion. We agree that structural DRR measures can sometimes create a false sense of security and may inadvertently reduce preparedness for residual risks. To reflect this insight, we added a brief note in Section 3.4 acknowledging that levees and similar structural defenses may foster complacency if not coupled with sustained awareness and preparedness efforts. | Added one sentence in Section 3.4 noting that structural defenses can create a false sense of safety and be counterproductive without complementary non-structural measures. | Section 3.4 |